# Cellulose Nanocrystals from Fibers of Macauba (*Acrocomia Aculeata*) and Gravata (*Bromelia Balansae*) from Brazilian Pantanal

**DOI:** 10.3390/polym11111785

**Published:** 2019-11-01

**Authors:** Ana Carolina Corrêa, Vitor Brait Carmona, José Alexandre Simão, Fabio Galvani, José Manoel Marconcini, Luiz Henrique Capparelli Mattoso

**Affiliations:** 1Nanotechnology National Laboratory for Agriculture (LNNA), Embrapa Instrumentation, P.O. Box 741, 13560-970 São Carlos, SP, Brazil; brait_carmona@hotmail.com (V.B.C.); alexandre_simao1@hotmail.com (J.A.S.); jose.marconcini@embrapa.br (J.M.M.); luiz.mattoso@embrapa.br (L.H.C.M.); 2Graduate Program in Materials Science and Engineering (PPGCEM), Federal University of São Carlos (UFSCar), Rod. Washington Luiz, Km 235, 13565-905 São Carlos, SP, Brazil; 3Embrapa Pantanal, P.O. Box 109, 79320-900 Corumbá, MS, Brazil; fabio.galvani@embrapa.br

**Keywords:** cellulose nanocrystals, pantanal fibers, gravata, macauba, characterization

## Abstract

Cellulose nanocrystals (CNC) were obtained from macauba and gravata fibers. Macauba (or Bocaiuva) is a palm tree found throughout most of Brazil and Gravata is an abundant kind of bromelia with 1–2m long leaves, found in Brazilian Pantanal and Cerrado. The raw fibers of both fibers were mercerized with NaOH solutions and bleached; they were then submitted to acid hydrolysis using H_2_SO_4_ at 45 °C, varying the hydrolysis time from 15 up to 75 min. The fibers were analyzed by X-ray diffraction (XRD), FTIR Spectroscopy, scanning electron microscopy (SEM) and thermal stability by thermogravimetric analysis (TG). XRD patterns did not present changes in the crystal structure of cellulose after mercerization, but it was observed a decrease of hemicellulose and lignin contents, and consequently an increase of cellulose content with the increase of NaOH solution concentration in the mercerization. After acid hydrolysis, the cellulose nanocrystals (CNC) were also analyzed by transmission electron microscopy (TEM) which showed an acicular or rod-like aspect and nanometric dimensions of CNC from both fibers, but the higher values of aspect ratio (L/D) were found on CNC obtained from gravata after 45 min of acid hydrolysis. The mercerization and subsequent bleaching of fibers influenced the crystallinity index and thermal stability of the resulting CNC, but their properties are mainly influenced by the hydrolysis time, i. e., there is an increase in crystallinity and thermal stability up to 45 min of hydrolysis, after this time, both properties decrease, probably due to the cellulose degradation by the sulfuric acid.

## 1. Introduction

Cellulose is a natural semi-crystalline polymer consisting of glucose repeating units bounded by β-1,4-glycosidic bonds; it can be produced and obtained from plants, bacteria and even some animals [1,2]. Cellulose nanocrystals (CNC) are rod-like and highly crystalline materials with diameters in the range of 5–70 nm and lenghs from 100nm up to a few microns. CNC can be obtained by the acid hydrolisis of high content semi-crystalline cellulose materials, such as vegetable fibers [3].

CNC are produced by the crystalline fraction exposure of cellulose fibers to strong acid hydrolysis. The chemical process starts with the removal of superficial polysaccharides followed by cleavage and destruction of amorphous fractions of cellulose. To end the reaction, the acid solution must be diluted and separated from the CNC by means of centrifugation and dialysis. Then, CNC should be redispersed by ultrasound producing stable suspensions. Both structure, properties and suspension behavior of CNC depends on concentrations and type of used acid, hydrolysis temperature and time and of intensity of ultrasound energy [3,4,5].

CNC can present different morphologies, dimensions and crystallinity depending on material source and the fiber pre-treatment, i.e., bleaching, mercerizing or using raw fibers, as well the time, temperature, acid type and concentration on acid hydrolysis [6].

CNC stable suspensions can be produced using concentrated solutions of H2SO4 and also HCl followed by sonication, and the first report of such a process was made by Ranby in 1951 [7]. Since then, CNC has been obtained via acid hydrolysis from various materials such as tunicates [8], microbial cellulose [9], Kraft pulp [10], microcrystalline cellulose [4] and vegetable fibers such as cotton [11,12], sugar-cane bagasse [13], sisal [14,15] and curaua [16].

The type of acid strongly affects superficial properties of CNC. CNC obtained from HCl hydrolysis tend to form aggregates [9,17], whereas those obtained from H_2_SO_4_ hydrolysis are stable and better dispersed due to the ester-sulphate electrostatic repulsion present on the crystal surfaces [18]. On the other hand, CNC morphology mainly depends on the cellulose source: CNC from tunicates and algae can have up to a few tens of microns in length, while vegetable fibers produce crystals with length in the order of hundreds of nanometers [3,8,18,19,20]. The type of acid plays important role on thermal stability of CNC as well. Better dispersed and surface charged CNC, obtained by H_2_SO_4_ hydrolysis, can be up to 95 °C less thermally stable than those obtained by HCl hydrolysis (~200 °C vs. 295 °C, respectively), while CNC obtained by an acid mixture present intermediate properties [16,21].

The development of CNC-reinforced nanocomposites still require study and advance because of CNC low thermal stability compared to processing temperature of engineering polymers, as well their hydrophobic character and inherent difficulty of dispersion in nonpolar matrices. However, CNC has already been used as reinforcement polymer matrix based on polysaccharides or proteins, resulting in increased mechanical strength and barrier properties without compromising their biodegradability [22,23].

Macauba (*Acrocomia aculeata*) and gravata (*Bromelia balansae*) present good properties and high content of cellulose [24,25], showing potential to be used as a source to obtain CNC. Macauba is an abundant palm tree found throughout most of Brazil, their leaves and nuts can be used as human and animal nutrition and for oil extraction, and their trunk can be used in civil construction. Gravata is an abundant kind of bromelia with 1–2m long leaves found in Pantanal and Brazilian Cerrado, their leaves are used to obtain natural fibers for handcrafts and ropes and their fruits (pulp and nuts) are appreciated by humans and animals [26]. Embrapa Pantanal manages a project with the Antonia Maria Coelho Community that aims to characterize the productive system of extracted products from macauba, like using its pulp and flour on vitamins, ice creams and breads, and its oil on biodiesel production. All these activities contribute to generate wealth for local communities and helps to promote sustainable development by encouraging projects whose uses raw material from renewable sources, adding values to these products. Images of gravata and macauba are shown in Figure 1. 

Thus, this is an exploratory study, proposing the use of byproducts (rachis, leaves and fruit pulp) of macauba and from gravata fibers, to obtain a cellulose rich material with potential to obtain cellulose nanocrystals (CNC), adding value to these materials and generating income for the local population. In this way, the aim of this study is the extraction and characterization of CNC from macauba and gravata. It will also be investigated the influence of different pre-treatment on raw fibers, using different concentrations of sodium hydroxide and peroxide solutions, and the influence of acid hydrolysis time on final properties of obtained CNC, such as thermal stability, crystallinity and morphology.

## 2. Experimental

### 2.1. Material and Methods

#### 2.1.1. Alkali Treatment of Gravata and Macauba Fibers

The CNCs from gravata and macauba was obtained from the fibers previously submitted to mercerization and/or bleaching, in order to achieve higher cellulose content and greater exposure of the cellulosic fibers. Gravata and Macauba raw fibers were gently supplied by Embrapa Pantanal (Corumbá, Brazil). The mercerization was carried out in sodium hydroxide (NaOH) solutions, and in order to evaluate the influence of the NaOH concentration on the alkaline solution for mercerization, 1%, 5% and 10% (*m*/*v*) of NaOH was applied. 20.0 g of fiber were placed in 300 mL of alkali solution at 60 °C, and submitted to mechanical stirring at 3000 rpm for 60 min.

It was also evaluated the influence of alkaline and peroxide concentration on the removal of non-cellulosic components from the fiber. Hydrogen peroxide was added to the alkaline solution in the amount that the final solution had the concentration determined in Table 1. Around 20 g of fibers were added to the hydrogen peroxide alkali solution at 60 °C and they were mechanical stirred for 60 min. The conditions of mixing, temperature and reaction time were kept constant to mercerization and bleaching processes. Furthermore, a bleaching treatment of each fiber was selected and repeated in the already bleached fiber under the same conditions of stirring, temperature and reaction time.

All samples were filtered through a filter paper, washed until neutralization and dried in an air circulation oven at 60 °C for 24 h. Table 1 shows the encoding of the gravata and macauba fibers as a function of pretreatment. 

#### 2.1.2. CNCs from Bleached Fibers Obtained by Acid Hydrolysis

The hydrolysis process was carried out based on previous studies, where conditions for other vegetable fibers such as cotton [12,21], curaua [16], sisal [14], eucalyptus [10], sugarcane bagasse [13], oil palm fibers [27], among others, were already established. From the bleached fibers, the acid hydrolysis was carried out adding 5 g of bleached fibers in 100 mL of 60% w/w H_2_SO_4_ solution at 45 °C under vigorous stirring for different reaction time: 15, 30, 45, 60 and 75 min. The resulting suspensions were centrifuged twice for 10 min at 10,000 rpm, the supernatant was discarded, the sediment was redispersed in distilled water and dialyzed in water until neutral pH. The suspension was sonicated for 5 min, frozen and stored. Dry CNCs were obtained by lyophilization process.

Table 2 presents the coding of the CNC samples, obtained from macauba and gravata from the once or twice bleached fibers and after different hydrolysis time.

### 2.2. Characterization

#### 2.2.1. X-Ray Diffracton

The X-ray diffractograms of raw, pretreated fibers and CNCs were obtained on an X-ray diffractometer (Shimadzu, XRD-6000, Tokyo, Japan), operating at 30 kV / 30 mA, CuKα radiation (λ = 1.5406 Å). The tests were conducted at room temperature and 2θ between 10 and 30° with a speed of 1°/min. The crystallinity indexes were calculated by deconvolution in peaks of the diffractograms, taking a Gaussian distribution function as crystalline and amorphous peaks. The OriginLab 8.0 software (OriginLab, Northampton, MA, USA) was used to estimate the crystallinity indexes (Ci) based on the areas under the crystalline peaks and amorphous halo of cellulose after correction of the baseline, according to the method of Oh et al. [28]. Equation 1 was used to estimate Ci:
(1)$$Ci\;\left(\%\right)=\;\left(1-\frac{A_{a}}{A_{t}}\right)\times 100$$
where, A_a_ is the area corresponding to the curve of the amorphous halo, and A_t_ is the sum of the areas of all crystalline and amorphous peaks.

#### 2.2.2. Infrared Spectroscopy (FTIR)

FT-IR analyzes of raw, pretreated fibers and CNCs were performed on a Perkin Elmer Spectrum 1000 spectrometer (Perkin Elmer, Waltham, MA, USA). The spectra were obtained with 64 scans in the wavelength region of 400 to 4000 cm^−1^ and resolution of 2 cm^−1^.

#### 2.2.3. Termogravimetric Analysis (TGA)

The thermal stability of raw, pretreated fibers and CNCs were evaluated by thermogravimetry using TGA Q500 (TA Instrument, New Castle, DE, USA) equipment under synthetic air atmosphere at flow rate of 60 mL/min; heating rate of 10 °C/min and temperature range from 25 to 600 °C. The onset temperature (*T*_onset_) was determined through the TG curve, as the intersection of the extrapolation line from the beginning of the thermal event with the tangent to the curve at the temperature of the maximum thermal degradation rate (obtained by the DTG curve) of the material.

#### 2.2.4. Scanning electron microscopy (SEM)

SEM micrographs of the outer surface of the in natura and pretreated fibers were obtained on a JEOL scanning electron microscope (JSM-6510 series, Jeol Ltd., Tokyo, Japan), operating at 10kV, and all samples were coated with gold.

#### 2.2.5. Transmission Electron Microscopy (TEM) 

Diluted suspensions of CNCs in water were prepared which were dispersed using Branson’s tip ultrasound (Branson Ultrasonics, Danbury, CT, USA), with a 1cm tip, at 50% power for 2 min. One drop of this suspension was placed on copper grids (400 mesh, Ted Pella - No. 01822, Pelco Inc., Fresno, CA, USA) and dried at room temperature. After 24 h, the samples were stained with 1.5% uranyl acetate solution, deposited on the grids. The analyzes were performed on a Tecnai ™ G2 F20 equipment (FEI Company, Hillsboro, OR, USA). The mean diameter as well as the mean length of the CNCs were calculated using ImageJ software (Bethesda, MD, USA) using at least 50 measurements in each dimension.

## 3. Results and Discussion

As the objective is to obtain cellulose nanocrystals, pre-treatments of mercerization and bleaching of the gravata and macauba fibers were carried out in order to extract the highest amount of hemicelluloses and lignin. The bleaching of both fibers was only carried out after prior characterization of the mercerized fibers, determining the concentration of NaOH and H_2_O_2_ in the alkaline peroxide solution used for bleaching. Figure 2 shows the gravata and macauba fibers after the mercerization and bleaching treatments.

As can be observed in Figure 2, there was a gradual whitening of the gravata and macauba fibers as the NaOH concentration was increased for each mercerization treatment. The use of alkaline peroxide in bleaching treatments has led to even more intense whitening, especially for macauba fibers. This can be considered as an indication of the efficiency of the treatments in the removal of compounds such as hemicellulose and lignin from the gravata and macauba fibers.

Fibers Ci can be related to their cellulose contents [29]. Thus, its determination by XRD is a good parameter for the selection of the appropriate concentration of NaOH for the mercerization of the vegetal fibers, as well as the alkaline peroxide solutions in the bleaching treatments. The diffraction profiles of raw, mercerized and bleached fibers from gravata and macauba are shown in Figure 3.

The XRD diffraction profiles of the fibers are similar to each other and show characteristic peaks of cellulose type I with diffraction peaks at 2θ = 15°, 17° and 22.7° [2]. The increase of NaOH concentration in solutions used for the mercerization and the increase of H_2_O_2_ concentration in the subsequent bleaching treatments caused a narrowing and definition of the crystalline peaks, indicating a removal of amorphous materials, such as hemicellulose and lignin [29].

However, even increasing NaOH concentration to 10 wt% (in the solution used for mercerization), no polymorphic changes of cellulose type I (raw fibers) to cellulose type II (after the mercerization process) were observed, maintaining the crystalline structure of raw cellulose fibers [30].

It was observed an increase of the Ci with the increase of the NaOH concentration for the mercerization treatments and also with the increase of the H_2_O_2_ concentration in the subsequent bleaching treatments, mainly for macauba samples, as presented in Table 3. And for both fibers, the repetition of the bleaching process provided an increase in the crystallinity of the fibers, which will be used to obtain the cellulose nanocrystals.

For the gravata fibers, it can be observed that the increase of the NaOH concentration in the mercerization was efficient, resulting in an increase of the Ci using solution with up to 5% NaOH (*m*/*v*), reaching Ci = 77% for gra_m2. And maintaining the concentrations of 5% NaOH (*m*/*v*) to prepare the alkaline peroxide solution for bleaching, it was observed an increase in Ci with the increase in the concentration of H_2_O_2_ up to 5% (*v*/*v*), reaching 82% of crystallinity for gra_b2. Repeating this bleaching process, Ci = 86% was determined for gra_b2_2x. But even using a 10 wt% NaOH solution for mercerization and increasing the concentration of H_2_O_2_ to 10 wt% in the alkaline peroxide for bleaching, it was not observed additional effects on the crystallinity of gravata fibers.

Regarding the macauba fibers, mercerization with a 10 wt% NaOH solution generated the highest Ci of 70.8% for mac_m3. Thus, maintaining the concentration of 10 wt% NaOH for the alkaline peroxide solutions used for bleaching, and increasing the concentration to 10% H_2_O_2_ (*v*/*v*) in the alkaline peroxide solution, a Ci of 77% was calculated for mac_b3; and after the repetition of this bleaching process, it was determined a Ci of 83% for mac_b3_2x. As macauba fibers present a higher non-cellulosic materials content than gravata fibers, like lignin and hemicellulose [24,25], it was expected to be necessary a more drastic treatment, in order to remove those substances.

In fact, what occurs is not an increase in the crystalline portion of the fibers but a removal of a greater part of the amorphous portion of the fibers, which can also be components of lower thermal stability. In this way, their removal result in an increase in thermal stability of the pre-treated fibers, as can be observed in Table 3. Thus, the mercerized fibers of gravata with 10wt% NaOH solution, presented an increase in thermal stability of up to 70 °C, reaching T_onset_ of around 314 °C. The subsequent bleaching treatments resulted in fibers with T_onset_ ranging from 302 to 311 °C.

Regarding the macauba fibers, the bleaching treatment resulted in an increase of up to 50^o^C in thermal stabilities (T_onset_ = 284 °C for mac_b3_2x). The T_onset_ of the remaining macauba fibers were determined in the range of 268 °C to 282 °C, much more thermally stable than fibers “in natura”, which shows the necessity of the pre-treatments of mercerization and/or bleaching of the vegetal fibers before the process of acid hydrolysis to obtain cellulose nanocrystals.

Figure 4 shows the TG and DTG curves of the gravata and macauba fibers and their thermal behavior.

In general, there is a first stage of mass loss occurring between room temperature and 160 °C, which is attributed to the presence of water absorbed or bound to the fibers. Between 160 °C and 500 °C occurs the loss of mass attributed to the thermal degradation of organic compounds, such as cellulose, hemicellulose and lignin. Among these, the hemicellulose is the component of lower thermal stability, which is degraded between 180 °C and 260 °C. The cellulose degrades between 240 and 350 °C, and the lignin between 280 °C and 400 °C. At higher temperatures, degradation byproducts of cellulose, hemicellulose, and lignin are degraded [31]. For both fibers gravata and macauba, it was observed a higher intensity and a shift for higher temperatures of the 1^st^ peak, after mercerizing and bleaching, which could be associated to the hemicellulose removal from the fibers and a consequent higher amount of cellulose on mercerized and bleached fibers [31].

Higher thermal stability can be observed for mercerized and bleached fibers, as well as narrowing and better definition of DTG peaks. This indicates that these treatments were efficient in the removal of compounds such as hemicellulose and lignin, as indicated by the XRD analyzes.

The characterization of vegetable fibers by FTIR is able to identify the presence of its main constituents: cellulose, hemicellulose and lignin. These components are composed mainly of alkanes and aromatic groups, as well as different functional groups containing oxygen, such as ester, ketone and alcohol [31,32], allowing monitoring the effect of chemical treatments on plant fibers.

Figure 5 shows the FTIR spectra of the raw, mercerized and bleached gravata and macauba fibers. Also indicated in the spectra are the absorption wave numbers of three characteristic bands: 1739 cm^−1^, relative to the stretching of C=O bonds of esters and carboxylic acids from hemicellulose; 1505 cm^−1^ relative to the bonds of C=C of benzene rings present in the lignin, and 1247 cm^−1^, referring to the stretching of C=O bonds of acetyl groups belonging to lignin [31,32]. But it can be observed in the FTIR spectra that the three bands mentioned were totally or partially suppressed (except the band at 1505 cm^−1^ for macauba), confirming the removal of hemicellulose and lignin from the gravata and macauba fibers.

In order to observe the fiber surfaces before and after the treatments, SEM analyzes were carried out and Figure 6 present SEM images for raw, mercerized and bleached fibers of gravata and macauba.

The surfaces of the gravata and macauba fibers before the treatments are covered by impurities specific to the fibers, such as waxes and other types of fatty acids. It can also be observed that the mercerizing treatments were able to partially remove these impurities, exposing the cellulosic fibers. With the bleaching treatments, these cleaning and fibrillar exposure processes were maximized. From this higher cellulosic fibers exposure, it is expected to provide a greater efficiency of the acid hydrolysis step, increasing the yield in obtaining CNCs with a shorter reaction time. 

Thus, it can be concluded that the most efficient treatments for the removal of non-cellulosic constituents were bleaching, performed once or twice. Thus, to continue the study of the production of cellulose nanocrystals from these fibers from Brazilian Pantanal, bleached fibers were used: gra_b2, gra_b2_2x, mac_b3 and mac_b3_2x.

In this way, CNC obtained from bleached fibers gra_b2 and gra_b2_2x will be called CNCg_1 and CNCg_2, respectively. And the CNC obtained from mac_b3 and mac_b3_2x will be called CNCm_1 and CNCm_2, respectively. And the termination corresponds to the hydrolysis time. 

Figure 7 shows the cellulose nanocrystals from gravata (CNCg) and from macauba (CNCm), obtained via acid hydrolysis at different times of hydrolysis and after freeze-drying.

The obtained CNCs showed coloration in lighter and darker shades of brown. Higher hydrolysis time resulted on darker CNCg unlike CNCm, wich remained with almost the same shade of brown. This darkening of CNCg may be related to some level of cellulose degradation caused by acid hydrolysis, since gravata cellulose fibrils were physically more exposed than those of macauba, as it can be observed by SEM images of pre-treated fibers. In addition, due to the higher exposure of the fibrils, there may have been a higher sulphonation of the CNCg with a longer hydrolysis time [33]. 

The morphology of the obtained CNCs was investigated by TEM analyzis, and the micrographs of the CNCg and CNCm samples are presented in Figure 8 and Figure 9, respectively.

From the TEM images it was possible to observe individual CNCs, revealing their acicular forms at the nanoscale, very similar to CNCs obtained from other plant sources such as cotton, ramie, wheat, curauá, sisal, among others [14,16].

Lengths and diameters of the CNC were determined from TEM images, as well as their aspect ratio (L/D), which are presented in Table 4.

From the average values of CNC lengths and diameters, it can be observed that there is a relationship between the CNC dimensions and the hydrolysis time. As the hydrolysis time increases, the length and diameter of the CNC are reduced [27]. It can also be observed that the number of bleaches that the fiber underwent prior to the step of hydrolysis interferes in the CNC dimensions only when these were obtained in shorter hydrolysis times (15 min, for example). Even at longer hydrolysis time, the number of bleaches in the fiber does not significantly change the average lengths and diameters of the CNC, as is the case, for example, of CNCg_1_75 and CNCg_2_75.

From the measurements of length and diameter of the CNC, it was possible to calculate their aspect ratios (L/D), an important parameter in relation to the use of them as reinforcement for polymer matrices, for example [34,35]. The L/D calculated values were in the range of 8 to 19, and the CNCg_2_45 presented the highest value. These values are in the range of values determined for other plant sources, such as sisal, curaua and cotton [12,14,16].

Regarding the crystalline structures of the obtained CNC, XRD analyzes were performed and Figure 10 present the XRD patterns for CNCg and CNCm, respectively.

The XRD profiles of CNCg and CNCm are similar to each other, showing characteristic peaks of cellulose type I with diffraction peaks at 2θ = 15°, 17° and 22.7° [2]. In addition, CNC showed the same characteristic peaks as their source fibers, demonstrating that the performed acid hydrolysis was not able to modify the cellulose crystal structure of the original fiber. On the other hand, the hydrolysis with strong acid has the capacity to remove non-cellulosic components of the vegetal fibers and also part of the amorphous fraction of cellulose, remaining the highly crystalline CNC.

The crystallinity indices of the CNC were determined by peak deconvolution of their XRD patterns and the determined values are given in Table 4. After acid hydrolysis there was an increase in Ci compared to those from bleached fibers. However, after longer hydrolysis times than 60 min the Ci values were reduced, indicating a possible destruction of the cellulose crystals. For both sources, gravata and macauba, the highest values of Ci were found around 45 min of acid hydrolysis.

The thermal behavior of the CNC was evaluated by thermogravimetry, and the TG/DTG curves of CNCg and CNCm are presented in Figure 11.

The CNCs showed an initial mass loss up to approximately 120 °C, associated with water evaporation. Then, a stable plateau is made up to approximately 200 °C, where the thermooxidative degradation of the cellulose is initiated. The thermal degradation of the CNC starts at temperatures 50 °C to 80 °C lower than their respective bleached fibers. This phenomenon is associated with the incorporation of sulfate groups in the structure of cellulose, which exert a catalytic effect on the thermal degradation reaction of cellulose [36]. In this way, CNC obtained by the acid hydrolysis using sulfuric acid degrade at temperatures lower than its original vegetable fibers. Thus, the longer the acid hydrolysis using sulfuric acid, the higher the level of cellulose sulfation, and the lower the thermal stability of the CNC [16]. The reduction on thermal stability of CNC compared to those of treated fibers, caused by the incorporation of sulphate groups on cellulose surface, was also expressed on DTG peaks as a low density left shoulder (around 220 °C), particularly on CNCg_1_75 and CNCg_2_75.

The *T*_onset_ of the CNC were determined and are presented in Table 4. The CNCg presented higher thermal stabilities than the CNCm, comparing the same hydrolysis time and the same number of bleaching. The CNCg *T*_onset_ were in the range of 235 °C and 276 °C, whereas for the CNCm they varied between 195 °C and 237 °C. The superior thermally stable behavior of gravata in comparison to macauba was observed for the CNC as well, as it occurred with its raw and pretreated fibers. Thus, CNCg_2_45 presented the highest L/D and thermal stability, being suitable for incorporation into polymer matrices.

## 4. Conclusions

Cellulose nanocrystals were extracted from macauba and gravata fibers successfully for the first time. TEM analyzes confirmed the obtaining of structures with nanometric dimensions and with acicular formats. The increase in hydrolysis time resulted in both the decrease in lengths (between 580 and 190 nm) and diameters (between 69 and 15 nm) of CNCg and CNCm.

The crystallinity indices of the CNCs began to present reductions after 45 min of hydrolysis, indicating that longer times of acid hydrolysis degrade the cellulosic crystals of CNCg and CNCm.

The thermal stabilities of CNC were between 195 °C and 260 °C, but longer hydrolysis time reduced CNC thermal stability. CNCg were more thermally stable than CNCm. These thermal stabilities indicated the potential of application of these CNC in polymer matrices, such as TPS/PCL, for the development of biodegradable nanocomposites.

Aiming the development of new bionanocomposites, CNC from gravatá, specifically CNCg_2_45, showed higher thermal stability and L/D, which are important parameters to provide wider processing windows and improved mechanical properties to bionanocomposites.

## Figures and Tables

**Figure 1 polymers-11-01785-f001:**
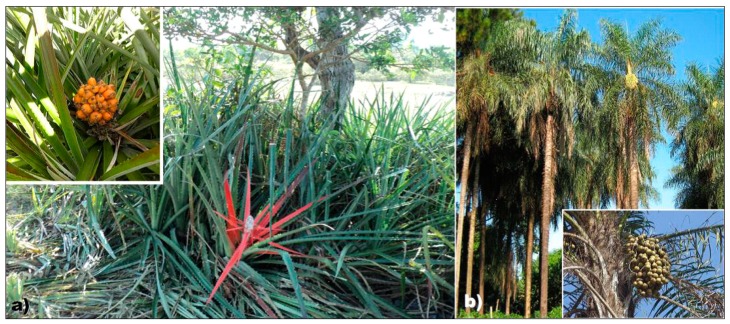
Images of (**a**) gravata and (**b**) macauba, with their respective fruits in detail.

**Figure 2 polymers-11-01785-f002:**
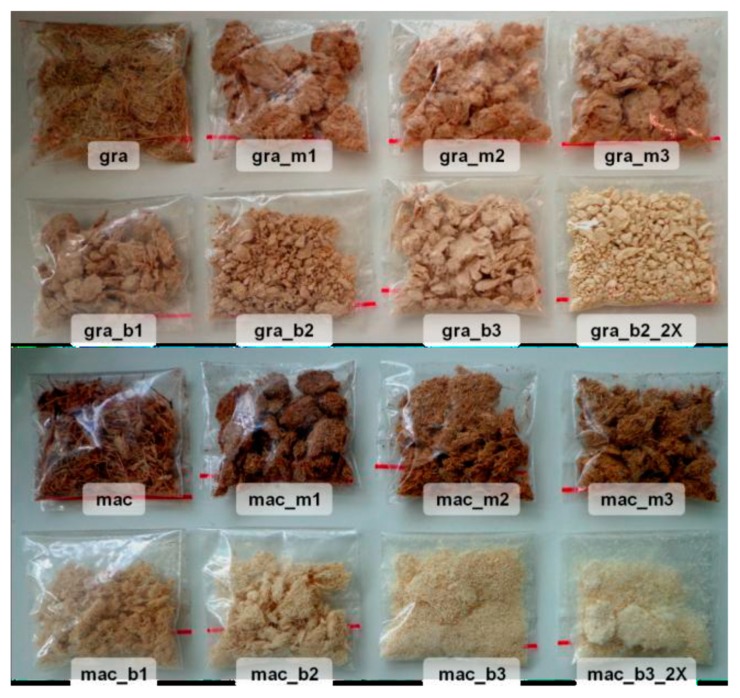
Images of raw fibers of gravata and macauba, and mercerized and bleached fibers, with their respective coding.

**Figure 3 polymers-11-01785-f003:**
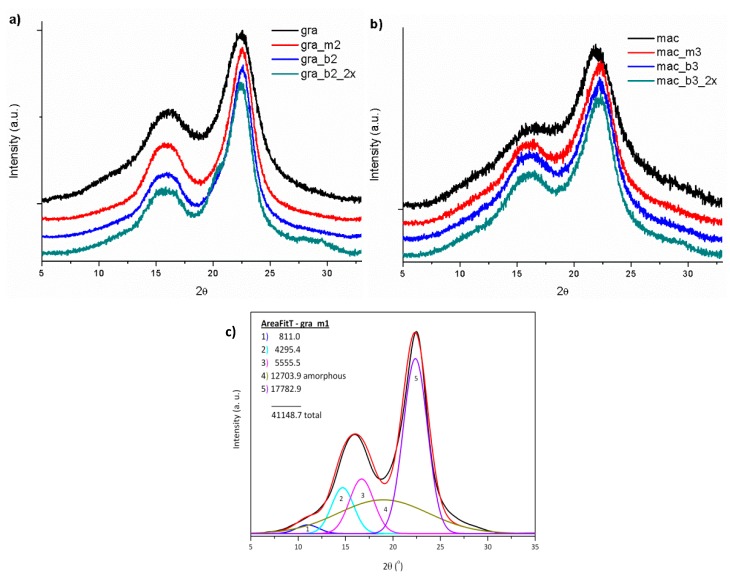
XRD profiles of raw, mercerized and bleached fibers from (**a**) gravata and (**b**) macauba, and (**c**) is an example of deconvoluted peaks for mercerized gravata fibers.

**Figure 4 polymers-11-01785-f004:**
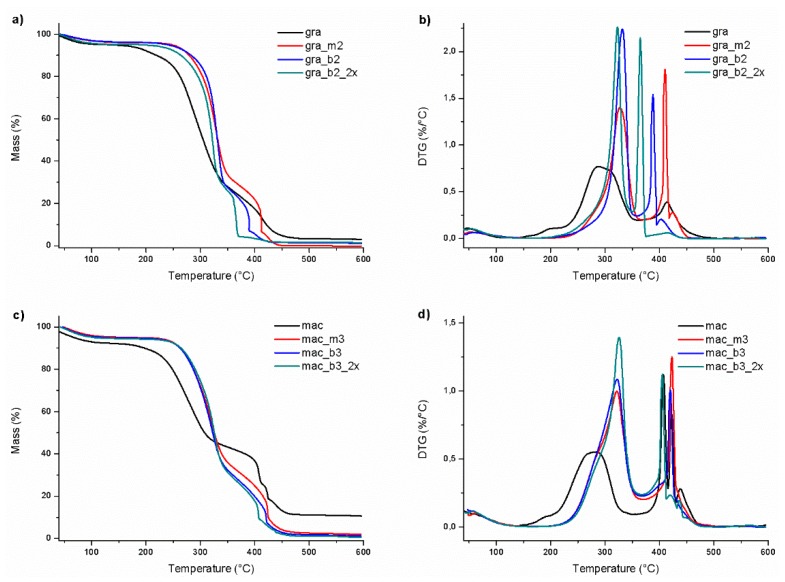
Thermograms **(a)** TG and (**b**) DTG curves of gravata raw, mercerized and bleached fibers, (**c)** TG and (**d**) DTG curves of macauba raw, mercerized and bleached fibers, in synthetic air atmosphere and heating rate of 10 °C/min.

**Figure 5 polymers-11-01785-f005:**
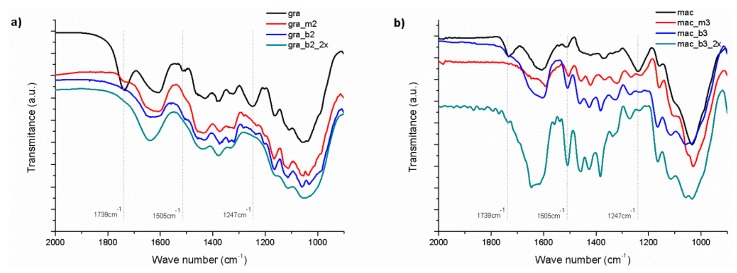
FTIR spectra of (**a**) gravata and (**b**) macauba: raw, mercerized and bleached fibers.

**Figure 6 polymers-11-01785-f006:**
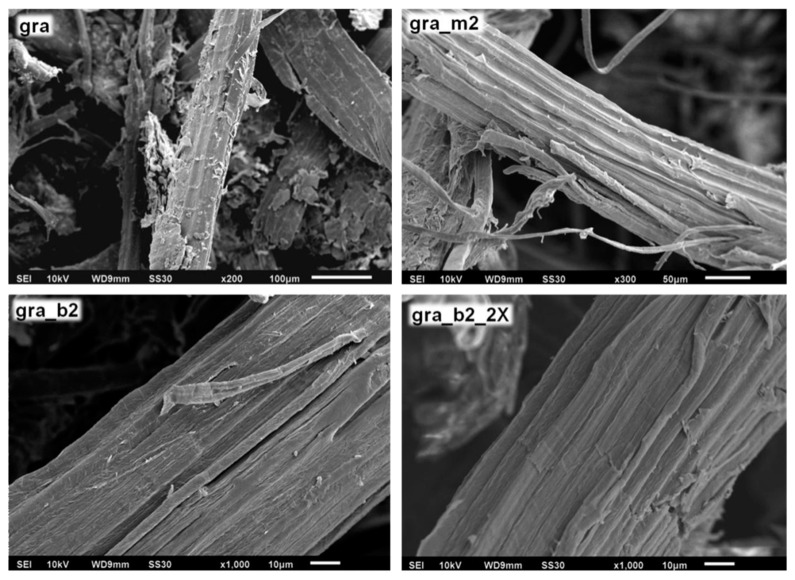
SEM micrographs of the surfaces of raw, mercerized and bleached fibers of gravata (gra) and macauba (mac).

**Figure 7 polymers-11-01785-f007:**
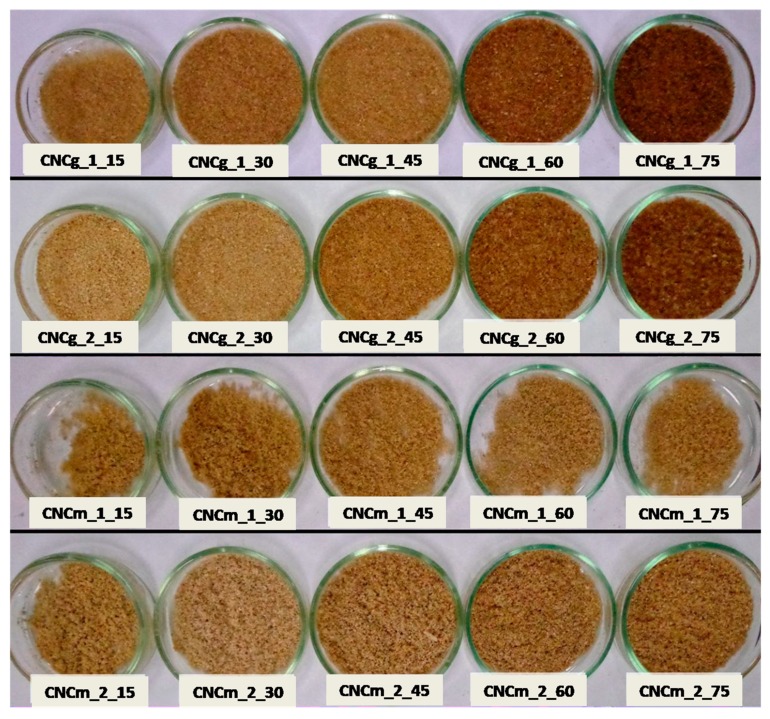
Visual aspect of CNCs from gravata and macauba fibers, after different times of acid hydrolysis and freeze-drying, with their respective encodings.

**Figure 8 polymers-11-01785-f008:**
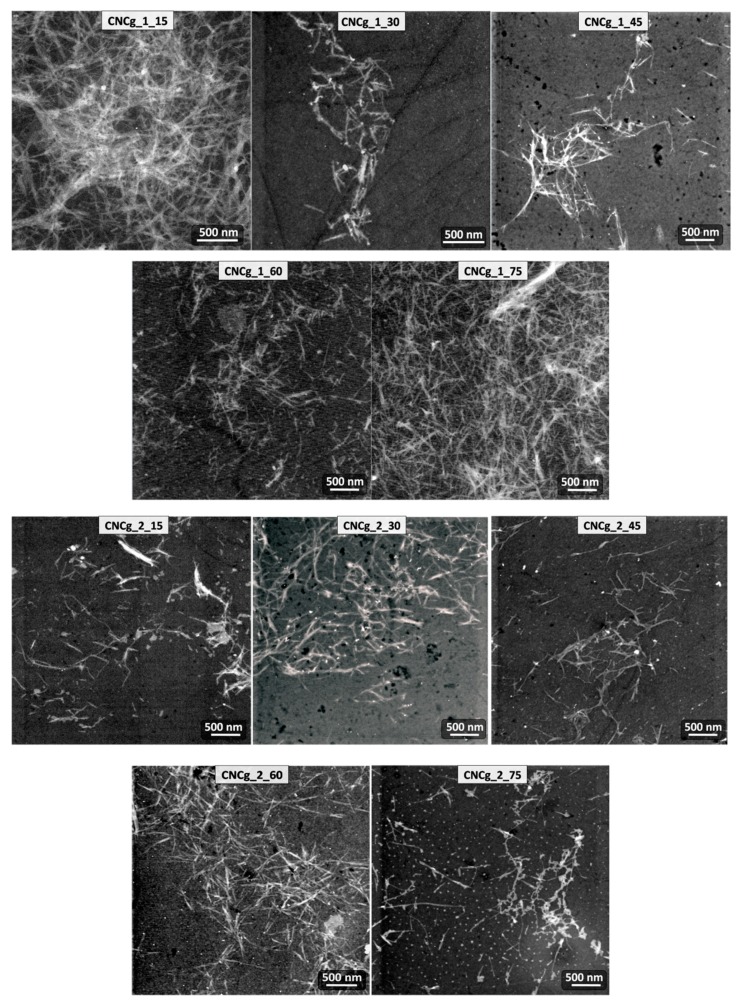
TEM micrographs of the CNCg samples with their respective encodings.

**Figure 9 polymers-11-01785-f009:**
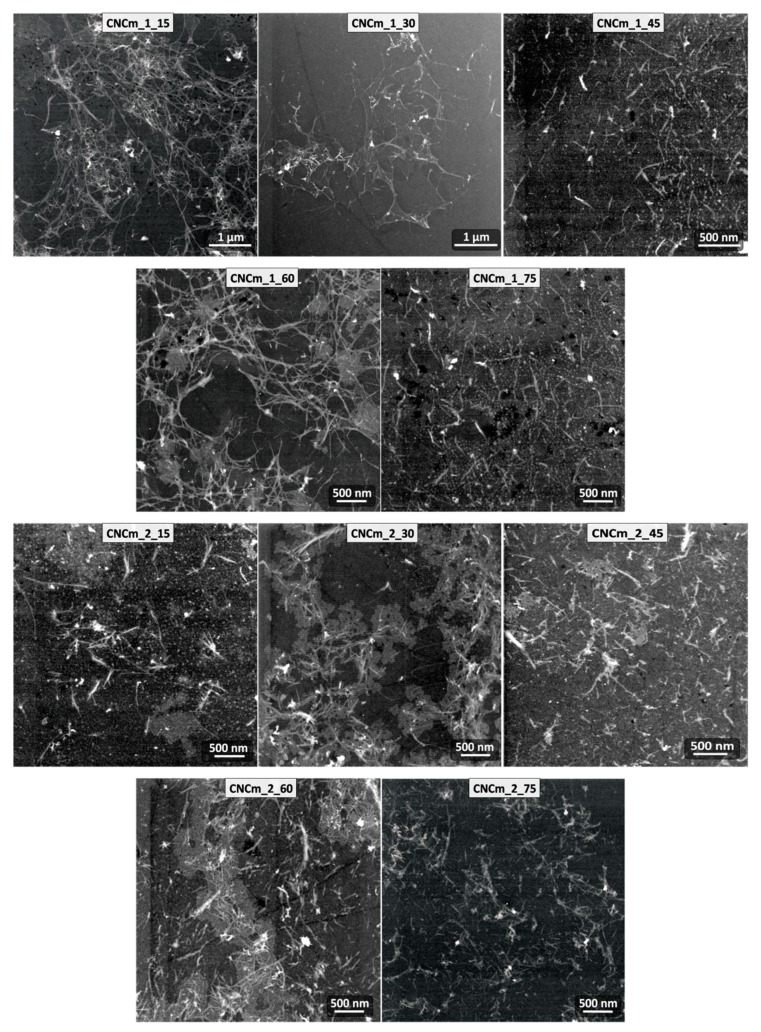
TEM micrographs of the CNCm samples with their respective encodings.

**Figure 10 polymers-11-01785-f010:**
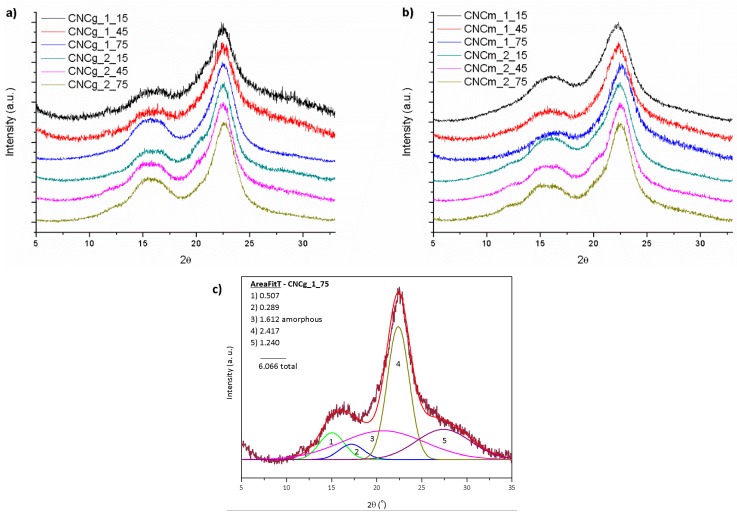
XRD profiles for the CNC obtained from (**a**) 1× and 2× bleached gravata fiber, (**b**) 1× and 2× bleached macauba fiber and (**c**) an example of deconvoluted peaks of CNCg_1_75.

**Figure 11 polymers-11-01785-f011:**
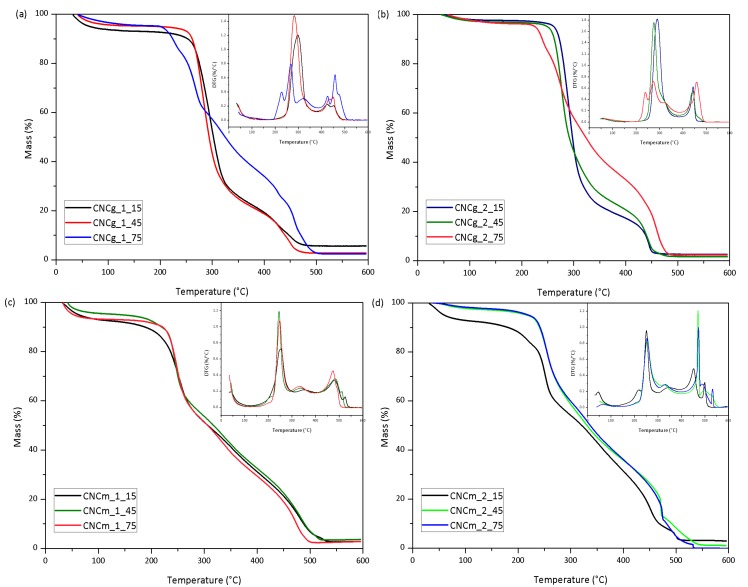
TG and DTG curves of CNCg obtained from (**a**) 1× and (**b**) 2× bleached gravata fiber; and CNCm obtained from (**c**) 1× and (**d**) 2× bleached macauba fiber; in synthetic air atmosphere and a heating rate of 10 °C/min.

**Table 1 polymers-11-01785-t001:** Gravata and macauba fiber samples under different mercerization and bleaching treatments.

Sample	Pretreatment	[NaOH] (*m*/*v*)	[H_2_O_2_] (*v*/*v*)
**gra**	None	-	-
**gra_m1**	Mercerization	1%	-
**gra_m2**	Mercerization	5%	-
**gra_m** **3**	Mercerization	10%	-
**gra_b1**	Bleaching	5%	1%
**gra_b2**	Bleaching	5%	5%
**gra_b3**	Bleaching	5%	10%
**gra_b2_2X**	Bleaching (2X)	5%	5%
**mac**	None	-	-
**mac_m1**	Mercerization	1%	-
**mac_m2**	Mercerization	5%	-
**mac_m3**	Mercerization	10%	-
**mac_b1**	Bleaching	10%	1%
**mac_b2**	Bleaching	10%	5%
**mac_b3**	Bleaching	10%	10%
**mac_b3_2X**	Bleaching (2X)	10%	10%

**Table 2 polymers-11-01785-t002:** CNC samples obtained from gravata and macauba, under different times of acid hydrolysis at 45 °C and from the once or twice bleached fibers.

Sample	Fiber	Hydrolysis Time (min)	Sample	Fiber	Hydrolysis Time (min)
**CNCg_1_15**	gra_b2	15	CNCm_1_15	mac_b3	15
**CNCg_1_30**	gra_b2	30	CNCm_1_30	mac_b3	30
**CNCg_1_45**	gra_b2	45	CNCm_1_45	mac_b3	45
**CNCg_1_60**	gra_b2	60	CNCm_1_60	mac_b3	60
**CNCg_1_75**	gra_b2	75	CNCm_1_75	mac_b3	75
**CNCg_2_15**	gra_b2_2X	15	CNCm_2_15	mac_b3_2X	15
**CNCg_2_30**	gra_b2_2X	30	CNCm_2_30	mac_b3_2X	30
**CNCg_2_45**	gra_b2_2X	45	CNCm_2_45	mac_b3_2X	45
**CNCg_2_60**	gra_b2_2X	60	CNCm_2_60	mac_b3_2X	60
**CNCg_2_75**	gra_b2_2X	75	CNCm_2_75	mac_b3_2X	75

**Table 3 polymers-11-01785-t003:** *T_onset_*, residues content and Ci of gravata and macauba raw, mercerized and bleached fibers.

Sample	*T_onset_* (°C)	*Residues (%)*	*Ci* (%)
**gra**	244	3.3	58
**gra_m1**	276	2.3	69
**gra_m2**	304	2.1	77
**gra_m3**	314	1.1	77
**gra_b1**	307	1.3	81
**gra_b2**	311	1.4	82
**gra_b3**	311	1.3	79
**gra_b2_2X**	302	1.3	86
**mac**	233	10.9	55
**mac _m1**	268	2.6	65
**mac _m2**	279	2.3	65
**mac _m3**	282	2.3	71
**mac _b1**	282	2.6	74
**mac _b2**	282	2.3	76
**mac _b3**	278	2.3	77
**mac _b3_2X**	284	1.1	83

**Table 4 polymers-11-01785-t004:** Values of Ci, T_onset_ and dimensions of CNC.

Sample	*Ci* (%)	*T*_onset_ (°C)	Length * (nm)	Diameter * (nm)	L/D
**CNCg_1_15**	80	268.6	582 ± 143 ^a^	69 ± 22 ^a^	8 ± 3
**CNCg_1_30**	80	259.2	322 ± 93 ^b^	30 ± 8 ^b^	10 ± 4
**CNCg_1_45**	82	264.2	345 ± 102 ^b^	32 ± 10 ^b^	11 ± 4
**CNCg_1_60**	76	250.3	289 ± 60 ^b,c^	22 ± 4 ^c^	13 ± 3
**CNCg_1_75**	73	235.1	257 ± 72 ^c,d^	19 ± 8 ^c^	14 ± 7
**CNCg_2_15**	87	276.5	373 ± 86 ^b,e^	36 ± 8 ^b^	10 ± 3
**CNCg_2_30**	87	258.5	386 ± 90 ^b,e^	29 ± 7 ^b^	13 ± 4
**CNCg_2_45**	88	263.5	383 ± 91 ^b,e^	20 ± 4 ^c^	19 ± 6
**CNCg_2_60**	84	249.0	282 ± 70 ^b,c,d^	19 ± 4 ^c^	15 ± 5
**CNCg_2_75**	82	240.6	217 ± 84 ^c,d^	24 ± 8 ^c^	9 ± 5
**CNCm_1_15**	78	230.5	499 ± 100 ^a^	36 ± 7 ^a^	14 ± 4
**CNCm_1_30**	79	230.2	462 ± 126 ^a^	34 ± 12 ^a^	13 ± 6
**CNCm_1_45**	81	231.0	428 ± 83 ^a,b^	34 ± 8 ^a^	13 ± 4
**CNCm_1_60**	83	227.8	359 ± 94 ^c^	30 ± 8 ^a^	12 ± 5
**CNCm_1_75**	80	224.6	231 ± 70 ^d^	22 ± 4 ^b^	10 ± 4
**CNCm_2_15**	83	234.7	410 ± 99 ^a,b,c^	40 ± 12 ^c^	10 ± 4
**CNCm_2_30**	83	237.7	377 ± 87 ^a,b,c^	34 ± 7 ^a^	11 ± 4
**CNCm_2_45**	86	237.7	304 ± 82 ^e^	27 ± 8 ^a^	11 ± 4
**CNCm_2_60**	80	230.6	238 ± 50 ^d^	22 ± 5 ^b^	11 ± 3
**CNCm_2_75**	77	194.8	171 ± 41 ^f^	15 ± 4 ^c^	11 ± 4

***** Values are averages of 50 measurements. Means accompanied by the same letter in the same column do not differ statistically according to the Tukey test.

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
