# Peer review of "Cellulose Nanocrystals from Fibers of Macauba (Acrocomia Aculeata) and Gravata (Bromelia Balansae) from Brazilian Pantanal"

_polymers, 2019, doi:10.3390/polym11111785_

Round 1

Reviewer 1 Report

Corrêa and co-workers herein report a method to synthesize cellulose nanocrystals (CNC) from macauba and gravata fibers. The effect of NaOH and H2O2 concentration on the mercerization and bleaching treatments were carefully studied. They also varied the time of acid hydrolysis to optimize the crystallinity and thermal stabilities of CNCs. This paper is well organized, and the experiments are carried out with care. Hence, this reviewer suggests an acceptance to Polymers after minor revisions.

Comments:

In figure 3, the peak of gra_b2_2x shifted in Figure a, while the peak of mac shifted in Figure b. What do these shifts mean? Page 7, line 198, while the authors claimed that Ic increased with the increasing NaOH concentration and H2O2 concentration, table 3 didn’t support it. For example, Ic of gra_m3 was smaller than gra_m2, and Error bars should be provided for Table 3. There was some typo that needed to be checked. For example, page 12, line 290, “MET analyzes”; line 291, it should be “Figure 8 and 9”. Page 14, line 320, this reviewer found that L/D increased with the increasing acidic hydrolysis time for CNCg, while decreased for CNCm_1. The L/D od CNCm_2 didn’t change with the acidic hydrolysis time. The authors need to explain this difference. There were too many profiles in each figure of Figure 11, making it not readable.

Author Response

In figure 3, the peak of gra_b2_2x shifted in Figure a, while the peak of mac shifted in Figure b. What do these shifts mean?

R. These small displacements in the gra_b2_2x and mac samples may be due to the moisture absorbed by the fibers prior to XRD characterization.

Page 7, line 198, while the authors claimed that Ic increased with the increasing NaOH concentration and H2O2 concentration, table 3 didn’t support it. For example, Ic of gra_m3 was smaller than gra_m2, and Error bars should be provided for Table 3.

R. Since Ic calculations were performed from the curve obtained by XRD from each sample, it is not possible to have a standard deviation, as there is only one curve for each sample, to observe the behavior of the fibers after each mercerization / bleaching process. And the data in table 3, referred to Ic, were summarized to whole numbers in order to decrease the accuracy of the adopted method.

There was some typo that needed to be checked. For example, page 12, line 290, “MET analyzes”; line 291, it should be “Figure 8 and 9”.

R: it was checked and corrected

Page 14, line 320, this reviewer found that L/D increased with the increasing acidic hydrolysis time for CNCg, while decreased for CNCm_1. The L/D of CNCm_2 didn’t change with the acidic hydrolysis time. The authors need to explain this difference.

R: length and diameter decrease with acid hydrolysis time, but this reduction may not be in the same proportion, so in some cases, as in the CNCg samples, the reduction in diameter was more significant than length, increasing the aspect ratio. This was not the case of the CNCm samples.

There were too many profiles in each figure of Figure 11, making it not readable.

R: the graphs in figure 11 were remade.

The author thanks for all suggestions.

Reviewer 2 Report

This article describes the production and characterization of cellulose nanocrystals (CNC) from macauba and gravata fibers. To the best of my knowledge the production of CNC using these sources is new and besides the interesting proposition of conferring value to these raw fibers, the work was well planned and all steps of production and characterization were well performed and evaluated using a number of techniques, e.g., TG-DTG, microscopy analysis using SEM and MEV, FTIR and XRD. The results were discussed and compared to the appropriate literature. Additionally, the authors have demonstrated correlations between the CNC properties and the parameters varied during the overall process, both those used for the pre-treatment, in which the concentrations of both alkali  and  H2O2  were varied, and also the reaction time. The results are new and the conclusions are sound and provide new perspectives for the production of CNC from natural fibers and I would recommend its publication after minor revision.

Remarks

In my opinion, a paragraph (or comment in the conclusion section) would be appropriate to compare the sources  for obtaining the CNC. Which one would be more appropriate taking into account the results obtained?

Page 2, lines 59 and 68 Typos:  repulsion and study instead of  “repulse” and “studied”

Table 1 must be corrected, bleaching instead of “breaching”.

Table 2 should be reorganized and reaction conditions for CNCs from macauba and gravata could be displayed side by side in two columns. 

Page 7, line 190: regarding the interpretation given for XRD profiles. the “diffraction peaks at 2theta 15, 17 and 22,7”, (three peaks?)  the peak at 15o is not visible in the XRD,  could the authors indicate it in the figure?

Page 11. Lines 288-289: A reference would be recommendable to support the statement “In addition, due to the higher exposure of the  fibrils, there may have been a higher sulphonation of the CNCg with a longer hydrolysis time”.

Page 17, lines 329-330 : Once more,  authors indicate three peaks  in XRD profiles (15, 17 and 22.7) and they are not easily seen in Fig. 10.

Author Response

This article describes the production and characterization of cellulose nanocrystals (CNC) from macauba and gravata fibers. To the best of my knowledge the production of CNC using these sources is new and besides the interesting proposition of conferring value to these raw fibers, the work was well planned and all steps of production and characterization were well performed and evaluated using a number of techniques, e.g., TG-DTG, microscopy analysis using SEM and MEV, FTIR and XRD. The results were discussed and compared to the appropriate literature. Additionally, the authors have demonstrated correlations between the CNC properties and the parameters varied during the overall process, both those used for the pre-treatment, in which the concentrations of both alkali  and  H2O2  were varied, and also the reaction time. The results are new and the conclusions are sound and provide new perspectives for the production of CNC from natural fibers and I would recommend its publication after minor revision.

 Remarks

 In my opinion, a paragraph (or comment in the conclusion section) would be appropriate to compare the sources  for obtaining the CNC. Which one would be more appropriate taking into account the results obtained?

R. Aiming the development of new bio nanocomposites, CNC from gravatá, specifically CNCg_2_45, showed higher values of Tonset and L/D, wich are important parameters to provide wider processing windows and improved mechanical properties.

Page 2, lines 59 and 68 Typos:  repulsion and study instead of  “repulse” and “studied”

R. it was correct in the manuscript.

Table 1 must be corrected, bleaching instead of “breaching”.

R. it was correct in the table 1.

Table 2 should be reorganized and reaction conditions for CNCs from macauba and gravata could be displayed side by side in two columns. 

R. table 2 was reorganized.

Page 7, line 190: regarding the interpretation given for XRD profiles. the “diffraction peaks at 2theta 15, 17 and 22,7”, (three peaks?)  the peak at 15o is not visible in the XRD,  could the authors indicate it in the figure?

R. there are three peaks in both profiles, as can be observed by the deconvolution, and an example graph was included in figure 3

Page 11. Lines 288-289: A reference would be recommendable to support the statement “In addition, due to the higher exposure of the  fibrils, there may have been a higher sulphonation of the CNCg with a longer hydrolysis time”.

R. A reference was added to this statement (Jarvis et al. 2018)

Page 17, lines 329-330 : Once more,  authors indicate three peaks  in XRD profiles (15, 17 and 22.7) and they are not easily seen in Fig. 10.

R. An additional graph was added to figure 10, in order to show the deconvolution of the diffractogram.

Reviewer 3 Report

The manuscript entitled “Cellulose Nanocrystals from Fibers of Macauba (Acrocomia aculeata) and Gravata (Bromelia balansae) from Brazilian Pantanal” regards the extraction and characterizations of cellulose nanocrystals (CNC) from macauba and gravata fibers. The raw fibers of both fibers were mercerized with NaOH solutions and bleached; they were then submitted to acid hydrolysis using H2SO4 at 45 °C, varying the hydrolysis time from 15 up to 75 min. The authors selected different characterizations to analyze the obtained materials starting to the raw fibers and the cellulosic materials after the different chemical processes.

The manuscript is well written and largely commented. I suggest the publication of this articles in Polymers journal but some specific comments should be introduced to improve the quality of the manuscript.

Specific comments

Introduction-paragraph 1: the authors are explain the novelty introduced with this research activity respect to the literature. The extraction of cellulose nanocrystals and the different time applied during the chemical procedure was largely investigated and proposed.  

References: K. B. R. Teodoro, E. d. M. Teixeira, A. C. Corrêa, A. de Campos, J. M. Marconcini, and L. H. C. Mattoso, Whiskers de fibra de sisal obtidos sob diferentes condições de hidrólise ácida: efeito do tempo e da temperatura de extração. Polímeros 21, 280–285 (2011). 25. X. M. Dong, J. F. Revol, and D. G. Gray, Effect of micro­crystallite preparation conditions on the formation of colloid crystals of cellulose. Cellulose (Dordrecht, Neth.) 5, 19 (1998). F. Luzi, E. Fortunati, D. Puglia, R. Petrucci, J.M. Kenny and L. Torre. Modulation of Acid Hydrolysis Reaction Time for the Extraction of Cellulose Nanocrystals from Posidonia oceanica Leaves JJournal of Renewable Materials, 4, 190-198(9), (2016).   Material and Methods-paragraph 2.1: The authors are invited to insert explain better the procedure applied during the pretreatment, specifically, they are invited to write the purification and the peroxide treatment used to obtain the cellulosic fibres. Result and discussion- paragraph 3-line 290, 302and 305: the authors are invited to replace the word MET with TEM. Line 312-313: The authors are invited to insert some references to support the following sentence:

As the hydrolysis time increases, the length and diameter of the CNC are reduced. Line 343: Figure 11 and not Figure 12. Please be careful to the numbering of the figures. Figure 11: The authors are invited to comment better the peaks of TGA. Conclusion: The authors are invited to underline the best treatment applied to obtain the CNC.

Author Response

Reviewer 3:

The manuscript entitled “Cellulose Nanocrystals from Fibers of Macauba (Acrocomia aculeata) and Gravata (Bromelia balansae) from Brazilian Pantanal” regards the extraction and characterizations of cellulose nanocrystals (CNC) from macauba and gravata fibers. The raw fibers of both fibers were mercerized with NaOH solutions and bleached; they were then submitted to acid hydrolysis using H2SO4 at 45 °C, varying the hydrolysis time from 15 up to 75 min. The authors selected different characterizations to analyze the obtained materials starting to the raw fibers and the cellulosic materials after the different chemical processes.

The manuscript is well written and largely commented. I suggest the publication of this articles in Polymers journal but some specific comments should be introduced to improve the quality of the manuscript.

Specific comments

Introduction-paragraph 1: the authors are explain the novelty introduced with this research activity respect to the literature. The extraction of cellulose nanocrystals and the different time applied during the chemical procedure was largely investigated and proposed.  

 References: K. B. R. Teodoro, E. d. M. Teixeira, A. C. Corrêa, A. de Campos, J. M. Marconcini, and L. H. C. Mattoso, Whiskers de fibra de sisal obtidos sob diferentes condições de hidrólise ácida: efeito do tempo e da temperatura de extração. Polímeros 21, 280–285 (2011).

M. Dong, J. F. Revol, and D. G. Gray, Effect of micro­crystallite preparation conditions on the formation of colloid crystals of cellulose. Cellulose (Dordrecht, Neth.) 5, 19 (1998).

Luzi, E. Fortunati, D. Puglia, R. Petrucci, J.M. Kenny and L. Torre. Modulation of Acid Hydrolysis Reaction Time for the Extraction of Cellulose Nanocrystals from Posidonia oceanica Leaves J Journal of Renewable Materials, 4, 190-198(9), (2016).

R. this item was added to the last paragraph of introduction

Material and Methods-paragraph 2.1: The authors are invited to insert explain better the procedure applied during the pretreatment, specifically, they are invited to write the purification and the peroxide treatment used to obtain the cellulosic fibres.

R. the item 2.1 was detailed explained.

Result and discussion- paragraph 3-line 290, 302 and 305: the authors are invited to replace the word MET with TEM.

R. thanks, they were replaced.

Line 312-313: The authors are invited to insert some references to support the following sentence: As the hydrolysis time increases, the length and diameter of the CNC are reduced.

R. a reference was inserted.

Line 343: Figure 11 and not Figure 12. Please be careful to the numbering of the figures.

R. figure numbers were corrected

Figure 11: The authors are invited to comment better the peaks of TGA.

R. Comments were inserted about DTG peaks. Thank you for suggesting.

Conclusion: The authors are invited to underline the best treatment applied to obtain the CNC.

R. It was inserted in conclusion section (as responded to Reviewer #2)